# CONTINUAL SUPERVISED ANOMALY DETECTION

## ABSTRACT

This paper proposes a continual-learning method for anomaly detection when a few labeled anomalies are available for training in addition to normal instances. Although several continual-learning methods have been proposed for anomaly detection, they have been dedicated to unsupervised anomaly detection, in which we can use only normal instances for training. However, few anomalies, which are valuable for constructing anomaly detectors, are often available in practice. In our continual-learning method, we use a hybrid model of a Variational AutoEncoder (VAE) and a binary classifier, and compute the anomaly score from the outputs of both models. The VAE is trained by minimizing the reconstruction errors of training data to detect unseen anomalous instances, and the binary classifier is trained to identify whether the input is a seen anomaly. Combining these two models enables us to efficiently detect both seen and unseen anomalies. Furthermore, the proposed method generates anomalous instances in addition to normal instances for generative replay to reduce the negative effects of catastrophic forgetting. In generative replay, anomalous instances are more difficult to generate than normal instances because few anomalous instances are available for training in anomaly detection. To overcome this problem, we formulate the generation of anomalous instances as an optimization problem, in which we find a latent vector of the VAE corresponding to anomalous instances, and generate anomalies by solving it using gradient descent. Our experimental results show that the proposed method is superior to anomaly detection methods using conventional continual learning.

## 1 INTRODUCTION

Anomaly detection is one of the key tasks in artificial intelligence (Chandola et al. (2009); Pang et al. (2021)). The goal of anomaly detection is to detect anomalous instances, called anomalies or outliers, from a given dataset. Anomaly detection has been used in various applications such as intrusion detection (Dokas et al. (2002)), defect detection (Tabernik et al. (2020)), fraud detection (Kou et al. (2004)), and medical care (Ukil et al. (2016)).

Many unsupervised anomaly detection methods have been proposed, including one-class Support Vector Machine (SVM) (Schölkopf et al. (2001)), isolation forests (Liu et al. (2008)), and Variational AutoEncoder (VAE) (Kawachi et al. (2018); An & Cho (2015); Xiao et al. (2020)). These methods detect anomalies by modeling the structure of normal instances. Although they have the advantage of not requiring anomalous instances, which are usually hard to obtain, they do not use any knowledge of anomalies, which would result in a high false positive rate. On the other hand, supervised anomaly detection methods (Pang et al. (2019; 2023); Zhou et al. (2022); Goyal et al. (2020)) have recently attracted attention for scenarios where few anomalies are available for training in addition to normal instances[1]. It has been reported that supervised learning can significantly improve anomaly detection performance thanks to the availability of anomalous instances for training. In fact, Han et al. (2022) have shown that most supervised anomaly detection methods outperform unsupervised methods even with only 1% of anomalous instances. It would be worthwhile to study supervised anomaly detection if anomaly detectors could be built with a sufficiently small number of anomalies that can be collected in real-world applications. Thus, this paper focuses on supervised anomaly detection methods.

---

[1]In this paper, semi-supervised anomaly detection, where we can use both unlabeled instances and labeled instances including normal and anomalous ones is also called supervised anomaly detection because unlabeled instances can be seen as normal instances due to the rarity of anomalies.

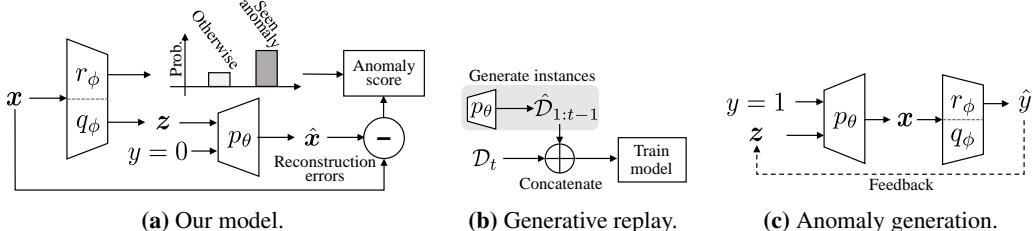

**Figure 1:** Proposed method: (a) Our model is a hybrid of a binary classifier and a Variational AutoEncoder (VAE). The binary classifier is modeled by the conditional probability distribution of a label $y$ given an instance $\boldsymbol{x}$, $r_\phi(y \mid \boldsymbol{x})$, where if $\boldsymbol{x}$ is a seen anomaly, then $y = 1$; otherwise, $y = 0$. The VAE consists of an encoder model $q_\phi(\boldsymbol{z} \mid \boldsymbol{x})$ and a decoder model $p_\theta(\boldsymbol{x} \mid \boldsymbol{z}, y)$, where $\boldsymbol{z}$ is a latent vector distributed according to the prior $p(\boldsymbol{z})$. The anomaly score is calculated by the outputs of both models, making detecting seen and unseen anomalies possible. (b) We generate a set of instances of the previous tasks, $\hat{\mathcal{D}}_{1:t-1}$, and combine $\hat{\mathcal{D}}_{1:t-1}$ and the current task dataset $\mathcal{D}_t$ for the training of the model, to prevent catastrophic forgetting. (c) Generating anomalies is difficult in generative replay because the number of anomalies available in training is very limited. Therefore, we propose formulating the generation of anomalies as an optimization problem, where the results of inputting the generated instances to the binary classifier are fed back to the latent vectors to search for the latent vectors corresponding to anomalies.

It has been reported that even in anomaly detection, using neural networks for representation learning can improve anomaly detection performance (Ruff et al. (2018; 2020)). Existing supervised anomaly detection methods based on neural networks primarily target static data (i.e., the distribution of observed data is stationary and does not vary over time). However, the distribution may change over time in real-world anomaly detection applications. For example, in monitoring network traffic, anomalous packets sent by an adversary may vary depending on the attacker's objectives and the type of attack. Normal packets may also vary depending on the content of the communication. In this way, such information may change over time, so anomaly detectors in such applications must be continuously trained and updated. Sequential updates of anomaly detectors may also be necessary if the model is difficult to train all at once due to the difficulty of retaining all previously collected data. For example, in the case of medical care, retaining all data collected in the past may be illegal or impossible due to patient privacy. In this case, the acquired data must be deleted after a certain period. In such cases, *continual learning*, in which the model is updated sequentially (Jung et al. (2016); Zenke et al. (2017); Prabhu et al. (2020); De Lange et al. (2022); Wang et al. (2023)), is one of the most promising ways to train neural networks. However, catastrophic forgetting is known as a problem that prevents continual learning (Kirkpatrick et al. (2017); Rudner et al. (2022); Guo et al. (2022b)). Catastrophic forgetting is a phenomenon in which after a neural network learns a new task, its performance on the previously learned tasks rapidly deteriorates. Although some continual learning methods for anomaly detection have been reported, their targets are unsupervised anomaly detection, and thus these methods are insufficient for supervised anomaly detection because they have difficulty making effective use of the labeled anomalies in training.

This paper proposes a continual-learning method for supervised anomaly detection. Figure 1 shows the overview of our proposed method. The contributions of this work are listed below.

- We propose a hybrid model of a VAE and a binary classifier for continual anomaly detection as shown in Fig. 1(a). The anomaly score is calculated from the reconstruction error of the VAE and the output of the binary classifier. This enables us to detect unknown (unseen) anomalies by using the reconstruction errors and known (seen) anomalies by using the binary classifier.

- For continually supervised anomaly detection, we present the generation method of anomalous instances using gradient descent to perform generative replay shown in Fig. 1(b). Our approach mitigates catastrophic forgetting by using a generative model to generate data from past tasks. Note that, in real-world scenarios, retaining even a small amount of past task data is not always possible due to privacy concerns. Among current countermeasures against catastrophic forgetting, generative replay is highly compatible with density estimation-based anomaly detection methods, which use a generative model to detect anomalies. In anomaly detection, the anomalous instances are difficult to model because

far fewer anomalies than normal instances are available for training. To overcome this problem, we formulate the generation of anomalies as an optimization problem, where we find a latent vector of the VAE corresponding to anomalous instances as shown in Fig. 1(c), and generate them by solving it using gradient descent. We can reduce the negative effects of catastrophic forgetting by using generated anomalous instances for generative replay.

- We experimentally demonstrate the effectiveness of the proposed method on some benchmarks for disjoint sequential tasks. The proposed generative replay method could be efficiently used to mitigate catastrophic forgetting, and we can avoid retraining the model for each task from scratch or storing some instances of the previous tasks. Our experimental results showed that our method outperformed other continual anomaly detection methods in detecting both seen and unseen anomalies.

## 2 RELATED WORK

**Anomaly Detection**   Current anomaly detection methods using neural networks can be divided into two categories in accordance with the number of anomalous instances available for training: unsupervised and supervised learning. In unsupervised learning-based methods, we use only normal instances to train an anomaly detection model. In supervised learning methods, we can use a small number of anomalies in addition to normal instances for training, and the anomaly detector is created by using both data. Unsupervised anomaly detection methods have the advantage of requiring only normal training data. Therefore, many unsupervised anomaly detection methods have been proposed, such as reconstruction-based methods (Gong et al. (2019); Perera et al. (2019); Somepalli et al. (2021); Sakurada & Yairi (2014)), classification-based methods (Ruff et al. (2018); Schölkopf et al. (2001)), and density estimation-based methods (Yoon et al. (2021); An & Cho (2015); Akcay et al. (2019); Zong et al. (2018)).

Although the unsupervised anomaly detection methods are the most feasible approaches because they do not require anomalies, their performance often degrades because they do not use prior knowledge of the anomalous data. In fact, previous studies Pang et al. (2019; 2023); Ruff et al. (2020); Han et al. (2022) have reported that supervised learning significantly improves anomaly detection performance even when very few anomalous instances are available for training. These methods use labeled instances to facilitate representation learning. For example, Pang et al. (2019); Ruff et al. (2020) train the model to increase the distance between normal and labeled anomalous instances in the latent vector space. Han et al. (2022) train an autoencoder model such that the reconstruction errors of the normal instances become small while those of the anomalous ones become large. To increase the reconstruction errors of anomalous instances, they use the deviation loss proposed by Pang et al. (2019). However, current approaches to supervised anomaly detection do not contain any mechanisms to update the model continually, which indicates that they would suffer from catastrophic forgetting. Therefore, we find a suitable mechanism for supervised anomaly detection in this paper.

Some unsupervised anomaly detection methods (Zheng et al. (2019); Sabokrou et al. (2018)) generate anomalous instances to improve anomaly detection performance similar to our method. However, they are not designed to generate the instances trained in the past, and assume that the data distribution is stationary. For this reason, they would be inappropriate for continual learning problem settings.

**Continual Learning**   Many methods for continual learning have been proposed, such as dynamic architecture-based methods (Rusu et al. (2022); Li & Hoiem (2018)), weight penalty-based methods (Egorov et al. (2021); Nguyen et al. (2018); Kirkpatrick et al. (2017); Guo et al. (2022a)), and generative replay-based methods (Shin et al. (2017); Varshney et al. (2022); Rao et al. (2019)). The dynamic architecture approach adds task-specific architecture (e.g., last layers) to a model for each task. The weight penalty approach introduces a regularization term for training new data to prevent significant changes from the parameters learned in the previous task. The generative replay prepares a generative model called a teacher model, and a student model (e.g., discriminative model). The teacher generative model is used for generating fake data that mimics former training instances. Then, the student model is trained on fake and new data.

The dynamic architecture approach has a problem that the size of the model architecture increases linearly with the number of tasks. In addition, if adding heads to the model in training, task identifiers are required during inference. Furthermore, the parameters of the added architecture are updated using only the dataset of a particular task, making it difficult to learn common concepts across tasks. The penalty weight approach has the advantage that we do not need to save any information (e.g., parameters, training data) from previous tasks, but a change in the weights is a poor proxy for the difference in model outputs (Benjamin et al. (2019)), which makes it difficult to avoid catastrophic forgetting fully. Generative replay has the disadvantage of requiring a teacher generative model. However, anomaly detection methods based on density estimation do not require a separate generative model because the generative model consisting of the anomaly detector can be used as a teacher model. In addition, generative replay can make it possible to avoid privacy risks and memory capacity limitations because we do not need to keep some instances of previous tasks for replay. Therefore, this paper focuses on anomaly detection methods based on generative models.

**Continual Learning for Anomaly Detection**   Some continual learning methods for anomaly detection have been presented and devoted to unsupervised anomaly detection (Wiewel & Yang (2019); Pezze et al. (2022); Frikha et al. (2021); Du et al. (2019)). Wiewel & Yang (2019) proposed a continual learning method for anomaly detection. The authors use a VAE as an anomaly detection model and show how to prevent catastrophic forgetting by using the decoder of the VAE trained in previous tasks as a teacher model for generative replay. Du et al. (2019) presented a continual anomaly detection method to improve the anomaly detection performance by using machine-unlearning techniques(Graves et al. (2021); Bourtoule et al. (2021)). They use a technique similar to Elastic Weight Consolidation (EWC) (Kirkpatrick et al. (2017)) to mitigate catastrophic forgetting of the model. (Pezze et al. (2022)) proposed a method to compress and store in memory the images needed for replay for continual learning. Frikha et al. (2021) proposed a meta-learning-based method for continual learning for unsupervised anomaly detection. Although these methods often work well for unsupervised anomaly detection, supervised anomaly detection, the main focus of this paper, is not covered by them. However, continual supervised anomaly detection can be more useful than unsupervised anomaly detection in some real-world applications, so we focus on it in this paper.

## 3   PROBLEM SETTINGS

The problem settings of current continual learning can roughly be divided into task-Incremental Learning (IL), domain-IL, and class-IL van de Ven & Tolias (2018). This paper focuses on domain-IL because there are two classes in classification (i.e., normal or anomalous instance) independently of tasks, and the task identifier is not given during inference.

Let $\mathcal{X} \subset \mathbb{R}^M$ be an input space, a subspace of the $M$-dimensional Euclidean space. Let $T$ be the number of tasks, and $N_t$ be the number of instances for the $t$-th task, where $t \in \{1, 2, \ldots, T\}$ is a task index. The $t$-th task dataset is defined by $\mathcal{D}_t = \{(\boldsymbol{x}_j^{(t)}, y_j^{(t)})\}_{j=1}^{N_t} \subset \mathcal{X} \times \{0, 1\}$, where $\boldsymbol{x}_j^{(t)}$ denotes the $j$-th instance of the $t$-th task, and $y_j^{(t)}$ denotes its label (i.e., if $\boldsymbol{x}_j^{(t)}$ is an anomaly, then $y_j^{(t)} = 1$, and otherwise, $y_j^{(t)} = 0$). Note that the distribution of the data points $(\boldsymbol{x}_j^{(t)}, y_j^{(t)})$ for dataset $\mathcal{D}_t$ can be different for each task (i.e., the data distribution of task $\mathcal{D}_t$ is denoted by $p_D^{(t)}(\boldsymbol{x}, y)$). We assume that there are far fewer anomalous instances than normal instances. In the training of the model, the datasets $\mathcal{D}_1, \mathcal{D}_2, \ldots, \mathcal{D}_T$ are given sequentially, and when training the $t$-th task, datasets other than the $t$-th task ($t'$-th task training dataset, where $t' \in \{1, \ldots, t-1, t+1, \ldots, T\}$) cannot be used for training the model. The goal of the prediction of the model trained on the $t$-th task is to classify unseen instances distributed in accordance with the distribution of the former tasks $p_D^{(t')}(\boldsymbol{x})$, where $t' \leq t$.[2]

---

[2]In this problem settings, test anomalies are drawn from the same distributions as the training data. See Appendix B.1 for experiments in which unseen anomalies appear during test.

## 4 PROPOSED METHOD

### 4.1 MODEL

The proposed anomaly detection model itself can be used independently of continual learning. Therefore, in this subsection, we describe the proposed model in the case of conventional normal learning, not in the context of continual learning.

Figure 1(a) shows the proposed model, a hybrid of a VAE and a binary classifier. Let $\mathcal{Z} \subset \mathbb{R}^K$ be a $K$-dimensional latent space, where $K$ is the dimension of latent variables $\boldsymbol{z} \in \mathcal{Z}$. Like VAE, the proposed model has an encoder $q_\phi(\boldsymbol{z} \mid \boldsymbol{x})$ and a decoder $p_\theta(\boldsymbol{x} \mid \boldsymbol{z}, y)$, where $\phi$ and $\theta$ are the parameters of neural networks. Similar to Conditional VAE (CVAE) (Kingma et al. (2014)), the proposed model models the distribution of $\boldsymbol{x} \in \mathcal{X}$ given the label $y$ by $p_\theta(\boldsymbol{x} \mid y) = \int p_\theta(\boldsymbol{x} \mid \boldsymbol{z}, y)p(\boldsymbol{z})d\boldsymbol{z}$, where $p(\boldsymbol{z})$ is the prior of $\boldsymbol{z}$.

The model parameters are determined by maximizing the evidence of the lower bound (ELBO) for an instance $(\boldsymbol{x}, y)$, which is given by

$$\log p_\theta(\boldsymbol{x} \mid y) = \log \mathbb{E}_{q_\phi(\boldsymbol{z}|\boldsymbol{x})} \frac{p_\theta(\boldsymbol{x} \mid \boldsymbol{z}, y)p(\boldsymbol{z})}{q_\phi(\boldsymbol{z} \mid \boldsymbol{x})} \geq \mathbb{E}_{q_\phi(\boldsymbol{z}|\boldsymbol{x})} \log \frac{p_\theta(\boldsymbol{x} \mid \boldsymbol{z}, y)p(\boldsymbol{z})}{q_\phi(\boldsymbol{z} \mid \boldsymbol{x})} = \mathcal{L}_{\text{ELBO}}(\boldsymbol{x}, y; \theta, \phi), \quad (1)$$

where $\mathbb{E}$ is an expectation operator, and $q_\phi(\boldsymbol{z} \mid \boldsymbol{x})$ is the conditional probability of $\boldsymbol{z}$ given $\boldsymbol{x}$. Eq. (1) can be written as

$$\mathcal{L}_{\text{ELBO}}(\boldsymbol{x}, y; \theta, \phi) = \mathbb{E}_{q_\phi(\boldsymbol{z}|\boldsymbol{x})} \log p_\theta(\boldsymbol{x} \mid \boldsymbol{z}, y) - D_{\text{KL}}(q_\phi(\boldsymbol{z} \mid \boldsymbol{x}) \parallel p(\boldsymbol{z})), \quad (2)$$

where $D_{\text{KL}}$ is KL divergence. The first term denotes the reconstruction errors, and the second term denotes the regularization term of the latent variable $\boldsymbol{z}$.

We can obtain the decoder model needed for generating data by maximizing Eq. (2) in terms of the parameters $\theta$ and $\phi$. Meanwhile, as mentioned in Subsection 4.3, generating anomalous instances for generative replay requires the binary classifier to discriminate whether a given instance $\boldsymbol{x}$ is a seen anomaly. In addition, this binary classifier is also used to calculate the anomaly score, as described in Subsection 4.2. Therefore, we introduce the conditional distribution $r_\phi(y \mid \boldsymbol{x})$ to create the binary classification model. Note that the two distributions $r_\phi(\boldsymbol{z} \mid \boldsymbol{x})$ and $q_\phi(y \mid \boldsymbol{x})$ are modeled by single neural network with the parameters $\phi$. Specifically, the neural network has $K + 1$-dimensional output, of which the 1-dimension corresponds to $y$ and $K$-dimension to $\boldsymbol{z}$. We train the model $r_\phi(y \mid \boldsymbol{x})$ by maximizing the log-likelihood $\log r_\phi(y \mid \boldsymbol{x})$ as well as ELBO in terms of $\phi$. Thus, the objective function of the proposed method is given by

$$L(\theta, \phi) := -\mathbb{E}_{p_D(\boldsymbol{x},y)}[\mathcal{L}_{\text{ELBO}}(\boldsymbol{x}, y; \theta, \phi) + \log r_\phi(y \mid \boldsymbol{x})], \quad (3)$$

where $p_D$ is the probability distribution of data.

### 4.2 ANOMALY SCORE FUNCTION

Given the data instance $\boldsymbol{x}$, the anomaly score in the proposed method is defined by

$$s_{\theta,\phi}(\boldsymbol{x}) := -\mathbb{E}_{q_\phi(\boldsymbol{z}|\boldsymbol{x})} \log p_\theta(\boldsymbol{x} \mid \boldsymbol{z}, y = 0) + \log r_\phi(y = 1 \mid \boldsymbol{x}), \quad (4)$$

The first term in Eq. (4) represents the reconstruction errors of $\boldsymbol{x}$, assuming that the instance $\boldsymbol{x}$ is normal. This is the same as conventional VAE-based anomaly detection, and allows us to detect of unseen anomalies because it is difficult to reconstruct untrained instances. The second term represents the log-likelihood of $y = 1$ given $\boldsymbol{x}$. If $\boldsymbol{x}$ is close to the seen anomalies given in training, this second term will become large. Therefore, the first and second terms correspond to unseen and seen anomalies, respectively. Note that we do not have to estimate the task identifier because our anomaly score does not require it.

### 4.3 GENERATION METHODOLOGY FOR GENERATIVE REPLAY

The proposed continual supervised anomaly detection uses the model described in Subsection 4.1 as a generative model to perform generative replay. We can easily generate normal instances by obtaining the conditional distribution $p_\theta(\boldsymbol{x} \mid \boldsymbol{z}, y = 0)$, where $\boldsymbol{z}$ is a latent variable sampled from the prior $p(\boldsymbol{z})$. On the other hand, generating anomalous instances from the conditional distribution

---

**Algorithm 1** Generating anomalous instances

---

**Require:** Parameters $\theta$ and $\phi$, number of anomalous instances to be generated $N_{\text{ano}}$
**Ensure:** Set of anomalous instances $\mathcal{D}_{\text{ano}}$
1: $\mathcal{D}_{\text{ano}} \leftarrow \emptyset$
2: **for** $i = 1 \ldots N_{\text{ano}}$ **do**
3:      $\boldsymbol{z}_i \sim p(\boldsymbol{z})$                                     $\triangleright$ Sampling $\boldsymbol{z}_i$ from the prior $p(\boldsymbol{z})$.
4:      **while** not converged **do**
5:          $\boldsymbol{z}_i \leftarrow \boldsymbol{z}_i - \eta \nabla_{\boldsymbol{z}} \left[ \mathbb{E}_{p_\theta(\boldsymbol{x}|\boldsymbol{z}_i, y=1)} \left[ \log r_\phi(\hat{y} = 1 \mid \boldsymbol{x}) \right] + \lambda \log p(\boldsymbol{z}_i) \right]$
6:      **end while**
7:      $\boldsymbol{x}_i \sim p_\theta(\boldsymbol{x} \mid \boldsymbol{z}_i, y=1)$
8:      $\mathcal{D}_{\text{ano}} \leftarrow \mathcal{D}_{\text{ano}} \cup \{\boldsymbol{x}_i\}$
9: **end for**

---

$p_\theta(\boldsymbol{x} \mid \boldsymbol{z}, y = 1)$, in the same way, does not work well. This is because there are far more normal instances than anomalous instances during training[3]. As a result, depending on the value of $\boldsymbol{z}$, normal instances may be generated even if $y = 1$ is conditioned. Therefore, to successfully sample anomalous instances, we formulate the generation of anomalies as an optimization problem.

The idea of generating anomalies is to find a latent vector $\boldsymbol{z}$ corresponding to anomalies by examining whether the instance generated from the latent vector $\boldsymbol{z}$ is classified as an anomaly by the trained binary classifier $r_\phi(y \mid \boldsymbol{x})$. To this end, we consider the following procedures.

1. Input a latent variable $\boldsymbol{z}$ sampled from the prior $p(\boldsymbol{z})$ and the label $y = 1$ to the decoder, and obtain the conditional distribution $p_\theta(\boldsymbol{x} \mid \boldsymbol{z}, y = 1)$.

2. Sample $\boldsymbol{x}$ from the conditional distribution $p_\theta(\boldsymbol{x} \mid \boldsymbol{z}, y = 1)$.

3. Obtain the conditional distribution $r_\phi(\hat{y} \mid \boldsymbol{x})$ by inputting the sampled instance $\boldsymbol{x}$ to the encoder, where $\hat{y}$ is the output of the binary classifier when inputting generated instance $\boldsymbol{x}$

Here, let us consider the conditional distribution of $\boldsymbol{z}$ given $y = 1$ and $\hat{y} = 1$, $p(\boldsymbol{z} \mid y = 1, \hat{y} = 1)$. If we find the latent variable $\boldsymbol{z}$ that maximizes this conditional distribution $p(\boldsymbol{z} \mid y = 1, \hat{y} = 1)$, then we can generate an instance $\boldsymbol{x}$ by sampling it from $p_\theta(\boldsymbol{x} \mid \boldsymbol{z}, y = 1)$. To calculate $p(\boldsymbol{z} \mid y = 1, \hat{y} = 1)$, we first rewrite it as

$$p(\boldsymbol{z} \mid y = 1, \hat{y} = 1) = \frac{p(\boldsymbol{z}, y = 1, \hat{y} = 1)}{p(y = 1, \hat{y} = 1)} = \int \frac{r_\phi(\hat{y} = 1 \mid \boldsymbol{x}) p_\theta(\boldsymbol{x} \mid \boldsymbol{z}, y = 1) p(\boldsymbol{z})}{p(\hat{y} = 1 \mid y = 1)} d\boldsymbol{x}. \quad (5)$$

Then, using Jensen's inequality, we have

$$\log p(\boldsymbol{z} \mid y = 1, \hat{y} = 1) \geq \mathbb{E}_{p_\theta(\boldsymbol{x}|\boldsymbol{z}, y=1)} \left[ \log r_\phi(\hat{y} = 1 \mid \boldsymbol{x}) \right] + \log p(\boldsymbol{z}) - \log p(y = 1, \hat{y} = 1). \quad (6)$$

Therefore, the objective function is given by $\mathbb{E}_{p_\theta(\boldsymbol{x}|\boldsymbol{z}, y=1)} \left[ \log r_\phi(\hat{y} = 1 \mid \boldsymbol{x}) \right] + \log p(\boldsymbol{z})$, and the latent vector to be obtained is

$$\hat{\boldsymbol{z}} = \arg\max_{\boldsymbol{z}} \mathbb{E}_{p_\theta(\boldsymbol{x}|\boldsymbol{z}, y=1)} \left[ \log r_\phi(\hat{y} = 1 \mid \boldsymbol{x}) \right] + \lambda \log p(\boldsymbol{z}), \quad (7)$$

where $\lambda \geq 0$ is a hyperparameter. The first term of Eq. (7) requires that the instance $\boldsymbol{x}$ generated from a given $\boldsymbol{z}$ be a seen anomalous instance. The second term requires that the given $\boldsymbol{z}$ follow the distribution of the prior $p(\boldsymbol{z})$. If $\boldsymbol{z}$ is obtained using only the first term, $\boldsymbol{z}$ may deviate significantly from the prior distribution. The regularization of the second term ensures that $\boldsymbol{z}$ falls within a meaningful region as a latent vector of VAE. However, if $p(\boldsymbol{z})$ is a Gaussian distribution with mean zero and the influence of the second term is too strong, the latent vector $\boldsymbol{z}$ becomes almost zero because the second term is $\log p(\boldsymbol{z}) = -\|\boldsymbol{z}\|^2/2$. To suppress this, we introduce the hyperparameter $\lambda$.

Algorithm 1 shows the procedure of generating an anomalous instance. The algorithm receives the trained parameters $\theta$ and $\phi$ and the number of anomalous instances to be generated $N_{\text{ano}}$ and returns

---

[3]Naturally, the number of anomalous instances should be small, so storing all anomalous instances of the past tasks may be one viable option. However, such an option is not chosen here because storing them as described in Section 1 is not always possible.

a set of generated anomalous instances $\mathcal{D}_{\text{ano}}$. The first line initializes the set $\mathcal{D}_{\text{ano}}$ to an empty set. In lines 4–6, we obtain the latent vector $z_i$ corresponding to the anomalous instances by using the gradient descent method. Here, $\eta$ is a hyperparameter. The gradient required for the gradient descent is obtained by using automatic differentiation. In line 7, $x_i$ is sampled from $p_\theta(x \mid z_i, y = 1)$ with the obtained $z_i$ and added to the set $\mathcal{D}_{\text{ano}}$.

### 4.4 OBJECTIVE FUNCTION FOR CONTINUAL ANOMALY DETECTION

This subsection describes the objective function for training the $t$-th task. Suppose that we have the parameters $\theta^{(t-1)}$ and $\phi^{(t-1)}$ trained on the previous tasks as mentioned in Section 3. Our method uses generative replay to mitigate catastrophic forgetting as shown in Fig. 1(b). Specifically, we first generate $N_{\text{no}}$ normal instances $\{x_{\text{no},i}\}_{i=1}^{N_{\text{no}}}$ and $N_{\text{ano}}$ anomalous instances $\{x_{\text{ano},j}\}_{j=1}^{N_{\text{ano}}}$ by using the method mentioned in Subsection 4.3 with the trained parameters $\theta^{(t-1)}$ and $\phi^{(t-1)}$. Then, we create a concatenated set $\mathcal{D}_{\text{gen}} := \mathcal{D}_t \cup \{(x_{\text{no},i}, y = 0)\}_{i=1}^{N_{\text{no}}} \cup \{(x_{\text{ano},j}, y = 1)\}_{j=1}^{N_{\text{ano}}}$. With the dataset $\mathcal{D}_{\text{gen}}$, we calculate and minimize the empirical risk of Eq. (3), which is defined by

$$L(\theta^{(t)}, \phi^{(t)}) \approx -\frac{1}{|\mathcal{D}_{\text{gen}}|} \sum_{(x,y) \in \mathcal{D}_{\text{gen}}} [\mathbb{E}_{q_{\phi^{(t)}}(z|x)}(\log p_{\theta^{(t)}}(x \mid z, y)) - \beta D_{\text{KL}}(q_{\phi^{(t)}}(z \mid x) \parallel p(z))$$
$$+ \log r_{\phi^{(t)}}(y \mid x)], \tag{8}$$

where $\beta \in \mathbb{R}_{\geq 0}$ is a hyperparameter as in $\beta$-VAE Higgins et al. (2017). By adjusting the value of the beta, the quality degradation of the generated data caused by training the latent vectors to follow a prior distribution can be minimized.

In actual training, we generate normal instances on a batch basis as in Wiewel & Yang (2019) instead of generating all previous normal instances before training. This eliminates the need to store generated normal instances for all previous tasks, and we need to save only the parameters that have been trained in the previous tasks. On the other hand, the number of anomalous instances to be generated would be sufficiently smaller than that of normal instances, so we generate all of them before training. The complete proposed training algorithm is described in Appendix D.

## 5 EXPERIMENTS

In this section, we demonstrate the effectiveness of our method on class-imbalanced datasets. We first describe the datasets and tasks used in this experiment and the comparison methods. Then, we show the anomaly detection performance of our methods and the comparison ones. Additional experiments to measure performance on unseen anomalies are described in Appendix B.1.

### 5.1 TASK

In our experiments, we used five datasets: MNIST (Deng (2012)), FMNIST (Xiao et al. (2017)), UNSW (Moustafa & Slay (2015)), bank (Moro et al. (2014)), and credit[4]. These datasets are commonly used in previous studies of continuous learning and anomaly detection(Pang et al. (2023); Wiewel & Yang (2019)). Since our method is a generic method for continuous anomaly detection, we validate the effectiveness of our method on a variety of datasets from different domains, including tabular datasets as well as image datasets. In the following, we explain how to create tasks from each real-world dataset. For MNIST and FMNIST, we divided the classes of each dataset in half; half of them are set to normal, and the others are set to anomalous (e.g., $\{0, 1, 2, 3, 4\}$ are normal, and $\{5, 6, 7, 8, 9\}$ are anomalous). Next, the classes were extracted one by one from the normal and anomalous data, and the pair of classes was made into a task (e.g., a task consists of the class pair $(0, 5)$). In this way, we created five tasks for MNIST and FMNIST. UNSW is a dataset of network packets and contains nine types of attacks. For UNSW, we created nine tasks by randomly and uniformly dividing the normal packets into nine subsets and assigning the attack packets of one of the attack types to each subset. The number of attack packets was reduced to 1/9 because the number of anomalous packets of some types of attacks (e.g., DOS attack) may be larger than the number of divided normal packets. The *bank* dataset is a dataset about direct marketing campaigns

---

[4]https://www.kaggle.com/datasets/mlg-ulb/creditcardfraud

of a Portuguese banking institution. Its objective variable is whether the customer applied for a term deposit as a result of the campaign. In this experiment, we assume that it is abnormal if the customer applies for the term deposit. In the dataset, the instances contain the customer's information, including their jobs, and therefore we split the data into 12 tasks according to them. The *credit* dataset includes credit card transactions made by European cardholders in September 2013. The dataset contains 284,807 transactions, of which 492 are fraudulent. In this experiment, we used 10% of the dataset as training data. To create the tasks, we divided the dataset equally into ten subsets in the time-series direction. The proportion of labeled anomalous instances in the training set to 5% for MNIST, FMNIST, and UNSW. To evaluate the trained models, we used the test dataset of all previously trained tasks and used the area under the receiver operating characteristic curve (AUC) as the evaluation metric. We ran ten experiments in each condition while changing the random seed and calculated the average AUC. The hyperparameters were determined by the AUC values for the validation set. The validation set was created by randomly dividing the training dataset by 30 percent. In this experiment, we normalize all instances $x$ to the interval $[0, 1]$.

## 5.2 COMPARISON METHODS

In this experiment, we used six famous anomaly detection methods for comparison: AutoEncoder (AE), Variational AutoEncoder (VAE), Binary Classifier (BC), Deep Semi-Supervised Anomaly Detection (Deep SAD) (Ruff et al. (2020)), Deviation Network (DevNet) (Pang et al. (2019)), and Boundary Guided Anomaly Detection (BGAD) (Yao et al. (2023)). Anomaly detection methods that do not use neural networks were not included in this experiment because the current continual learning methods are difficult to apply to them. AE and VAE are unsupervised anomaly detection methods using only normal instances, while BC, Deep SAD, DevNet, and BGAD are supervised anomaly detection methods that use anomalous instances in addition to normal instances for training. BC is a naive binary classifier, which is implemented by a neural network trained with a binary cross-entropy function. Deep SAD is an extension of Deep SVDD (Ruff et al. (2018)) for supervised anomaly detection (Ruff et al. (2020)). DevNet is a supervised anomaly detection method similar to Deep SAD. BGAD is a recent supervised anomaly detection method based on a normalizing flow. We used three popular methods for continual learning to combine with the above anomaly detection methods: Elastic Weight Consolidation (EWC) (Kirkpatrick et al. (2017)), A-GEM (Chaudhry et al. (2019)), and VAE-based Generative Replay (VAE-GR) (Wiewel & Yang (2019)). EWC, and A-GEM are applicable to any anomaly detection model, while the VAE-GR is for anomaly detection models with VAE. We used PyTorch and Avalanche (Lomonaco et al. (2021)), a library for continual learning, to implement these continual learning methods.

The hyperparameters of the model were selected by using the average validation AUC of the model after training on all tasks. Details of the experiments, including the network architecture of the proposed and comparative methods and candidate hyperparameters, are described in Appendix A.

## 5.3 RESULTS

Table 1 shows the average and standard deviation of the AUCs on all test datasets for the model after training on the final task. "LB" means that the results of the model trained only on the final task. Thus, the AUCs of all continual learning methods are expected to be higher than these values. Methods that were not significantly different in the t-test with $p = 0.05$ compared to the method with the highest averaged AUC are shown in bold.

First, we compare conventional unsupervised and supervised anomaly detection methods. It can be seen that the trends are completely different depending on the domain of the data. For the tabular datasets (i.e., UNSW, bank, and credit), BC, Deep SAD, DevNet, and BGAD, which use anomalous instances as well as normal instances during training, perform better than the unsupervised methods. Meanwhile, for the image datasets, we can see that both supervised and unsupervised methods perform similarly. This difference may be because the method used to create the tasks in this experiment produces larger differences in distribution between tasks for image datasets than for table datasets. Supervised methods outperform unsupervised methods because supervised methods can make better use of labeled anomalies in the training set. However, since the number of labeled anomalies is very small in anomaly detection, anomalies trained in previous tasks are likely to be forgotten compared to normal data. In particular, for image datasets, because of the large differ-

**Table 1:** Test AUCs [%] averaged over 10 runs after continually training on all tasks.

|          |        | FMNIST | MNIST | UNSW | bank | credit |
|----------|--------|--------|-------|------|------|--------|
|          | A-GEM  | 82.14±4.50 | 75.49±8.51 | 84.74±1.86 | 79.09±3.15 | **95.69±0.45** |
| AE       | EWC    | 64.84±9.01 | 70.37±8.69 | 85.06±1.84 | 78.71±2.20 | **95.60±0.65** |
|          | LB     | 60.05±8.41 | 61.42±9.52 | 80.72±2.17 | 67.76±2.58 | 94.55±0.63 |
|          | A-GEM  | 75.20±8.82 | 76.92±5.88 | **98.64±2.12** | **85.68±4.51** | **96.24±2.04** |
| BC       | EWC    | 65.89±10.30 | 69.49±5.70 | **98.60±2.16** | **86.36±2.71** | **96.59±1.92** |
|          | LB     | 58.65±8.27 | 60.91±5.87 | 97.10±3.44 | 73.86±7.15 | 95.62±4.09 |
|          | A-GEM  | 72.26±4.60 | 71.05±2.39 | **97.88±2.29** | **84.19±3.29** | **94.00±5.17** |
| Deep SAD | EWC    | 64.53±8.30 | 64.41±6.15 | **97.95±2.58** | **85.33±2.89** | 96.39±2.87 |
|          | LB     | 60.90±8.29 | 57.13±4.98 | 97.05±2.77 | 73.57±7.67 | 95.66±4.18 |
|          | A-GEM  | 66.14±8.76 | 76.56±10.43 | 81.11±0.38 | 75.95±1.04 | 94.62±0.64 |
| VAE      | EWC    | 60.45±10.29 | 61.10±9.63 | 81.13±0.61 | 76.35±0.99 | 94.62±0.63 |
|          | VAE-GR | 69.58±8.55 | 83.07±10.68 | 78.82±0.66 | 76.12±1.12 | 94.61±0.65 |
|          | LB     | 60.39±10.62 | 59.77±10.32 | 80.30±0.54 | 70.98±2.09 | 94.55±0.71 |
|          | A-GEM  | 66.39±8.90 | 68.69±6.04 | **90.45±11.41** | 73.08±5.62 | 92.41±4.58 |
| DevNet   | EWC    | 61.77±9.68 | 65.60±6.50 | **94.82±5.35** | 78.23±4.03 | **95.17±3.60** |
|          | LB     | 61.34±12.49 | 60.05±6.30 | 91.66±9.24 | 70.27±6.68 | 91.25±6.14 |
|          | A-GEM  | 78.01±4.98 | 77.85±6.15 | 96.52±2.22 | **84.41±2.05** | 95.52±0.89 |
| BGAD     | EWC    | 69.98±11.63 | 70.91±8.40 | 95.83±1.75 | **86.40±1.80** | **95.66±0.97** |
|          | LB     | 65.91±7.16 | 63.85±7.76 | 91.33±6.33 | 72.74±4.00 | 94.39±1.93 |
| Ours     |        | **88.23±2.56** | **95.46±1.64** | **98.15±2.21** | **86.75±2.93** | **96.12±1.22** |

**Table 2:** Averaged test AUC [%] when a component is removed from the proposed model.

|          | FMNIST | MNIST | UNSW | bank | credit |
|----------|--------|-------|------|------|--------|
| Ours     | 88.23±2.56 | 95.46±1.64 | 98.15±2.21 | 86.75±2.93 | 96.12±1.22 |
| w/o rec. | 82.88±4.64 | 87.87±4.13 | 96.60±0.70 | 86.18±1.89 | 96.04±1.25 |
| w/o bin. | 82.37±2.92 | 87.89±7.63 | 84.26±1.61 | 73.84±1.65 | 94.67±0.75 |
| w/o LVO  | 81.03±3.68 | 85.10±7.44 | 96.64±0.91 | 86.40±1.87 | 96.01±0.98 |

ences in distribution among tasks, anomalies of previous tasks are easily forgotten, which reduces the advantage of the supervised method. This is probably the reason why there was no difference in performance between unsupervised and supervised methods on the image datasets.

Next, we compare the results of the proposed method with those of the conventional methods. Table 1 shows that the proposed method is superior to all the combinations of conventional anomaly detection and continual learning methods in all cases. In addition, Table 1 also shows that the proposed method has relatively small standard deviations, which indicates that our method provides stable performance. These results indicate that the proposed generative replay method successfully mitigates catastrophic forgetting.

**Ablation study:** Table 2 shows the result of an ablation study to show that each of the components of our model plays an important role. "w/o rec." and "w/o bin." mean that the results without the reconstruction error and the log-probability of the binary classifier in our anomaly score (Eq. (4)). LVO in Table 2 stands for Latent Vector Optimization. In other words, "w/o LVO" means the results using generative replay without the proposed latent vector optimization (i.e., latent vectors $z$ sampled from the multivariate Gaussian distribution are directly used to generate anomalous instances). The table shows that the proposed model with all components is superior to the model without each component. Interestingly, relatively high AUCs were achieved even when the anomaly score was computed solely from reconstruction error. This is probably because our model can reconstruct only normal instances by conditioning on the label $y = 0$ when calculating the anomaly score in Eq. (4).

## 6 CONCLUSION

In this paper, we proposed a method of continual learning for supervised anomaly detection. The proposed model is a hybrid of a Variational AutoEncoder (VAE) and a binary classifier of anomalous instances, which can detect both unknown and known anomalies. Furthermore, by combining the generation of instances by VAE and the output of the binary classifier, we can generate not only normal instances but also anomalous instances. The proposed generation method for anomalous instances can be used to reduce the negative effects of catastrophic forgetting. We showed that the

proposed method outperformed various existing methods through experiments on five datasets. Although we used VAE as the generative model in this paper, future work will involve investigating the applicability of other density estimation methods such as autoregressive, flow-based, and diffusion models.

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

## A    Experimental Setup Details

In our paper, all experiments were conducted on a Linux workstation with two AMD EPYC 7543 32-Core processors, eight NVIDIA A30 GPUs, and 512 GB memory.

### A.1    Proposed Method

To implement the proposed method, we used a multivariate Gaussian distribution $\mathcal{N}(\mathbf{0}, \boldsymbol{I})$ for the prior distribution of $\boldsymbol{z}$. The L2 norm was used to compute the reconstruction error $\mathbb{E}_{q_\phi(\boldsymbol{z}|\boldsymbol{x})} \log p_\theta(\boldsymbol{x} \mid \boldsymbol{z}, y)$ in the error function shown in Eq. (2) because the L2 distance performed better than the binary cross-entropy in our experiments. The reconstruction error in the anomaly score was also calculated using the L2 distance. The $\boldsymbol{z}$ sampling for the reconstruction error calculation was done by the reparameterization trick (Kingma & Welling (2014)). The expected value was calculated by sampling a latent variable $\boldsymbol{z}$ multiple times and using a Monte Carlo method. The number of sampled instances for the Monte Carlo method was set to 1 for the training loss function and 100 for the anomaly score function. When generating anomalous instances by Algorithm 1, we set $\eta = 0.01$ and updated each $\boldsymbol{z}_i$ 1,000 times by gradient descent. The ratio of generated anomalous instances for the generative replay was always set to be 5%. Normal instances were generated for each batch, as described in Subsection 4.3. The probability distributions $q_\phi(\boldsymbol{z} \mid \boldsymbol{x})$ and $r_\phi(y \mid \boldsymbol{x})$ are modeled by a feed-forward neural network that has three hidden layers with 300 and 300 units from the input side for MNIST, UNSW, bank, credit and 1,024 and 1,024 for FMNIST. We use a leaky rectified linear unit (ReLU) function as the activation function. The output sizes of the encoder were 64, 64, 32, 32, and 16 for MNIST, FMNIST, UNSW, bank, and credit, respectively. Of the all dimensions of the output, one dimension was used to represent $y$, and the remaining dimensions were used to represent $\boldsymbol{z}$. We use the sigmoid function only for the one-dimensional output corresponding to $y$. The distribution $p_\theta$ was also modeled by another feed-forward neural network with three hidden layers with 300, 300, and 784 units from the input side for MNIST, UNSW, bank, and credit and 1024, 1024, and 784 units for FMNIST. Leaky ReLU was used for the activation function in the intermediate layers and the sigmoid function for the output layer. The proposed method was implemented using PyTorch (Paszke et al. (2019)).

### A.2    Comparison Methods

In our experiments, we compared the proposed method with four anomaly detection methods, and four continual learning methods. To implement the continual learning methods, we use Avalanche (Lomonaco et al. (2021)), which is an open source library for continual learning.

**Anomaly Detection Methods**

- **AE** is the autoencoder-based anomaly detection method (Sakurada & Yairi (2014)). We used the same network architecture as for the proposed method, except for the output function of the encoder. Specifically, we did not use the sigmoid function for its output because all the output dimensions of the encoder correspond to the latent vector $\boldsymbol{z}$ in the AE.

- **BC** is the binary classifier. We used the same network architecture as the encoder of the proposed method, except for the dimension of the output. Specifically, the number of output dimensions of the network architecture for the BC was one, and the sigmoid function was used for the output function.

- **VAE** is the Variational AutoEncoder. We used the same network architecture as for the AE. For VAE, we used the binary cross-entropy loss function to compute the reconstruction error as in (Wiewel & Yang (2019)).

- **Deep SAD** is an extension of Deep SVDD (Ruff et al. (2018)) for supervised anomaly detection (Ruff et al. (2020)). This method first trains the AE in advance. Then, the trained AE is used to transform the instances into latent vectors, and the encoder is trained such that the latent vectors of normal instances are close to the given center $c$, and the latent vectors of anomalous instances are further away from $c$. To train Deep SAD, AE needs to be trained in advance to determine the center $c$ in the latent space. In the continual learning problem setting, it is impossible to use the datasets of all tasks to determine $c$. Therefore, we trained

AE on the first task and calculated the center $c$ from the trained AE. In the experiment, $c$ determined in the first task was continuously used in the training of subsequent tasks. We used the same network architecture as for the AE.

- **DevNet** is a supervised anomaly detection method. Similar to Deep SAD, DevNet maps a given data point into a one-dimensional latent vector space and calculates the anomaly score based on the distance from the mean of the normal latent vectors. The normal latent vectors are sampled from a Gaussian distribution, and the previous study (Pang et al. (2019)) calculates the mean and standard deviation of the normal latent vectors using 5,000 points. We used the same network architecture as for the BC.

- **BGAD** is a supervised anomaly detection method, especially for the image domain (Yao et al. (2023)). BGAD can detect anomalies by converting input images into feature maps using a backbone model and then learning the distribution of the feature maps using normalizing flow. BGAD finds the boundary between normal and anomalous data in the latent vector space obtained by the normalizing flow, and uses this information for efficiently training using anomalies through boundary-guided semi-push-pull (BG-SPP) loss. In this paper, we did not use the backbone model because one of the goals of this paper is how to continually train models from scratch using multiple tasks consisting of a small number of data, and the availability of models trained on large datasets in advance, such as backbone models, is inconsistent with this goal. Therefore, the inputs of the normalizing flow are the input data, not the feature maps. The model architecture used for normalizing flow consists of eight coupling layers according to the GitHub implementation.[5]

**Continual Learning Methods**

- **EWC** stands for Elastic Weight Consolidation, a weight-penalty-based continual learning method. EWC computes the Fisher Information Matrix (FIM) from the loss function and introduces a regularization term using its diagonal components in order to suppress large changes from the parameters of the model trained in the previous tasks. Since the magnitude of the diagonal components of the FIM is determined in accordance with the importance of each dimension of the weights for the previous tasks, the regularization term based on the FIM can prevent significant changes in the dimensions of the weights that are important for the previous tasks.

- **VAE-GR** is a generative replay-based method for continual learning of unsupervised anomaly detection using VAE Wiewel & Yang (2019). This method uses the decoder of the VAE model in the anomaly detector to generate normal instances for generative replay.

- **A-GEM** stands for Averaged Gradient of Episodic Memory and is a faster version of GEM Chaudhry et al. (2019). First, A-GEM stores some instances of the previous tasks in episodic memory. Then, the stored instances are used to calculate the loss of the current model, and the gradient of back-propagation is modified so that the loss value of the current model evaluated using the episodic memory does not exceed the loss value of the model before training the current task.

## A.3 HYPERPARAMETERS

We selected hyperparameters using average validation AUC. For VAE and the proposed method, the regularization parameter $\beta$ was chosen from $\{0.1, 1, 10\}$. For the proposed method, the parameter $\lambda$ in the latent vector optimization was chosen from $\{0, 0.01, 0.1, 1\}$. For Deep SAD, the regularization parameter for the loss of anomalous instances $\eta$ was chosen from $\{0.1, 1, 10\}$. For BGAD, the hyperparameters $\beta$ and $\tau$ for BG-SPP loss were set to 0.05 and 0.1 according to its GitHub implementation. For EWC, the regularization parameter for the weight-penalty of continual learning was chosen from $\{10^{-2}, 10^{-1}, \ldots, 10^{2}\}$. For A-GEM, the number of instances per experience in episodic memory was set to 128, and the batch size for the calculation of gradients using the episodic memory was set to 32. The number of epochs for all methods was chosen from $\{10, 50, 100\}$, the batch size was set to 128, and we used the Adam optimizer (Kingma & Ba (2015)) with a learning rate of 0.001.

---

[5]https://github.com/xcyao00/BGAD

**Table 3:** Average test AUCs [%] of **seen and unseen anomalies** over 10 runs after continually training on half the tasks.

|  |  | FMNIST | MNIST | UNSW | bank | credit |
|---|---|---|---|---|---|---|
| AE | A-GEM | **84.23±6.82** | 78.18±13.90 | 83.15±2.23 | 79.28±2.65 | **95.53±0.63** |
|  | EWC | 70.08±10.30 | 70.21±13.72 | 83.52±2.21 | 79.75±2.95 | **95.66±0.56** |
|  | LB | 65.32±8.93 | 64.95±14.47 | 79.24±4.44 | 66.74±4.58 | 94.67±0.63 |
| BC | A-GEM | 73.56±7.02 | 75.41±10.64 | **94.74±3.88** | 86.29±2.93 | 96.16±1.50 |
|  | EWC | 60.68±13.22 | 72.41±9.59 | 89.53±7.55 | **86.55±2.61** | **96.18±1.34** |
|  | LB | 57.81±13.97 | 65.31±10.68 | 69.78±16.83 | 77.22±2.35 | 95.62±1.06 |
| Deep SAD | A-GEM | 75.04±4.99 | 73.69±6.01 | **92.16±4.43** | 82.58±6.38 | 95.54±0.96 |
|  | EWC | 65.99±8.47 | 65.25±4.76 | 92.24±2.97 | **82.99±8.11** | 94.98±1.38 |
|  | LB | 59.31±12.97 | 59.55±6.22 | 87.95±5.83 | 74.33±8.17 | 94.69±1.14 |
| VAE | A-GEM | 71.14±13.32 | 80.40±13.77 | 80.93±0.53 | 75.41±1.61 | 94.69±0.70 |
|  | EWC | 63.30±11.80 | 65.40±15.88 | 81.12±0.50 | 75.60±1.95 | 94.68±0.70 |
|  | VAE-GR | 73.55±10.42 | **87.16±9.54** | 79.20±1.15 | 76.93±1.74 | 94.56±0.76 |
|  | LB | 63.17±12.00 | 61.72±16.05 | 80.33±0.50 | 69.91±3.49 | 94.67±0.71 |
| DevNet | A-GEM | 60.20±12.62 | 66.29±13.96 | 87.79±5.43 | 75.18±4.85 | 92.75±3.76 |
|  | EWC | 62.27±10.12 | 71.18±9.89 | 89.20±3.52 | 77.44±6.20 | 93.72±1.85 |
|  | LB | 58.03±11.88 | 65.02±12.33 | 77.67±21.85 | 71.40±6.30 | 92.23±4.89 |
| BGAD | A-GEM | 78.67±10.79 | 80.45±10.83 | **94.69±3.78** | 82.68±2.71 | **95.79±0.96** |
|  | EWC | 66.15±11.43 | 71.52±12.98 | **93.70±3.53** | **85.77±1.85** | **95.85±0.93** |
|  | LB | 62.39±13.51 | 65.87±13.22 | 89.85±7.86 | 76.00±4.95 | 95.19±0.94 |
| Ours |  | **88.21±4.37** | **93.76±6.81** | **95.29±1.47** | **86.38±1.94** | **96.18±1.20** |

**Table 4:** Average test AUCs [%] of **unseen anomalies** over 10 runs after continually training on half the tasks.

|  |  | FMNIST | MNIST | UNSW | bank | credit |
|---|---|---|---|---|---|---|
| AE | A-GEM | **85.91±9.48** | **76.08±18.93** | 80.25±14.89 | 81.00±3.36 | **95.61±1.01** |
|  | EWC | 68.36±15.79 | 68.42±18.44 | 81.14±13.65 | 81.36±3.94 | **95.56±1.04** |
|  | LB | 61.81±13.12 | 64.51±17.97 | 77.94±13.36 | 69.01±6.36 | 94.72±1.37 |
| BC | A-GEM | 63.42±11.20 | 67.64±14.81 | **91.40±7.00** | 85.43±3.98 | **96.44±1.26** |
|  | EWC | 52.55±13.14 | 63.74±11.55 | **83.90±16.56** | 85.50±3.83 | **96.51±1.12** |
|  | LB | 48.57±17.48 | 57.67±15.28 | 70.16±25.13 | 76.98±3.73 | 95.90±0.75 |
| Deep SAD | A-GEM | 73.04±8.11 | 71.03±6.43 | **91.34±8.42** | 82.27±6.76 | 95.63±1.45 |
|  | EWC | 59.37±10.58 | 60.94±5.76 | **91.94±8.28** | 83.21±6.55 | 95.33±1.00 |
|  | LB | 55.37±17.62 | 58.17±8.49 | 85.35±13.18 | 75.41±8.25 | 94.82±1.48 |
| VAE | A-GEM | 69.86±16.02 | **77.89±23.43** | 76.57±17.67 | 78.08±3.54 | 94.74±1.17 |
|  | EWC | 59.19±17.83 | 63.65±22.34 | 76.74±17.43 | 78.32±3.71 | 94.73±1.16 |
|  | VAE-GR | 72.98±14.59 | **82.21±21.63** | 75.01±16.76 | 78.54±3.05 | 94.62±1.19 |
|  | LB | 59.15±18.25 | 60.74±20.54 | 75.58±18.32 | 71.95±4.96 | 94.70±1.04 |
| DevNet | A-GEM | 55.40±20.65 | 64.15±12.06 | **85.25±6.26** | 74.60±5.36 | 92.66±3.34 |
|  | EWC | 52.87±18.06 | 63.43±14.65 | 86.47±13.07 | 77.07±7.42 | 94.03±1.29 |
|  | LB | 51.82±19.11 | 55.22±15.28 | 82.73±21.77 | 71.35±6.62 | 92.53±5.69 |
| BGAD | A-GEM | 73.95±13.01 | 72.84±17.03 | **93.02±9.23** | 82.24±3.39 | **95.86±1.24** |
|  | EWC | 60.60±14.07 | 66.61±15.69 | **92.18±9.96** | 85.20±3.03 | **95.89±1.19** |
|  | LB | 59.05±13.06 | 60.31±14.99 | 90.15±14.34 | 76.37±5.07 | 95.35±1.25 |
| Ours |  | **85.91±9.46** | **88.42±16.00** | **94.35±3.03** | **85.94±2.91** | **96.38±1.41** |

## B ADDITIONAL EXPERIMENTAL RESULTS

### B.1 ROBUSTNESS AGAINST UNSEEN ANOMALIES

In this subsection, we conducted an experiment to confirm the effectiveness of the proposed method on unseen data. In this experiment, the way of creating tasks for training was the same as the problem settings mentioned in Section 5. On the other hand, when creating the test dataset for each task, we added the instances not only from the previously trained classes but also from all anomalous classes. This makes it possible to measure the detection performance of the proposed method on unseen anomalous instances since anomalies that have not been trained in the past will appear.

To measure performance when both unknown and known anomalous instances appear equally in the test dataset, Tables 3 and 4 show the test AUCs after training on half the tasks[6]. These tables show that the proposed method has consistently high performance in all tasks. Compared to Tables 3 and 1, VAE and AE perform better than the other conventional methods on the FMNIST and MNIST datasets, which may be due to the inclusion of unknown anomalies in the test data.

---

[6]If the number of total tasks $T$ was odd, the model trained on the $(T + 1)/2$th task was used.

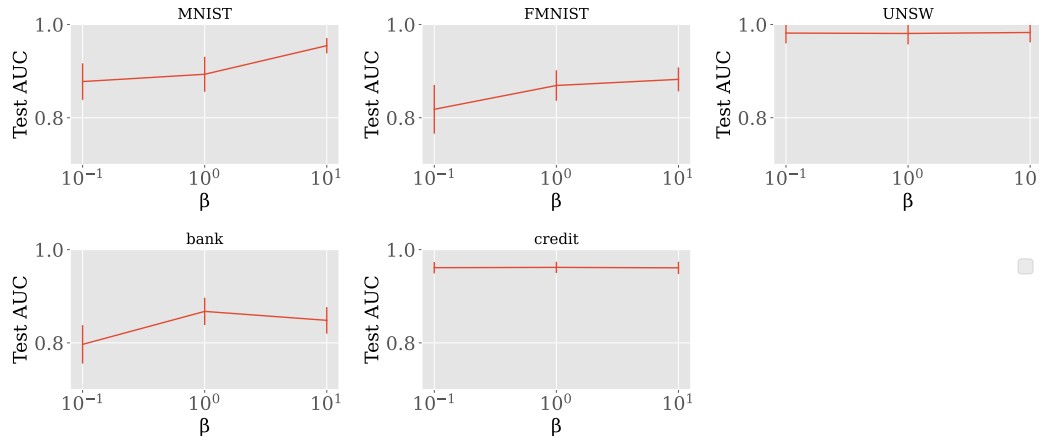

**Figure 2:** Test AUCs on the test dataset averaged over 10 runs after training on the final task **when $\beta$ changes**.

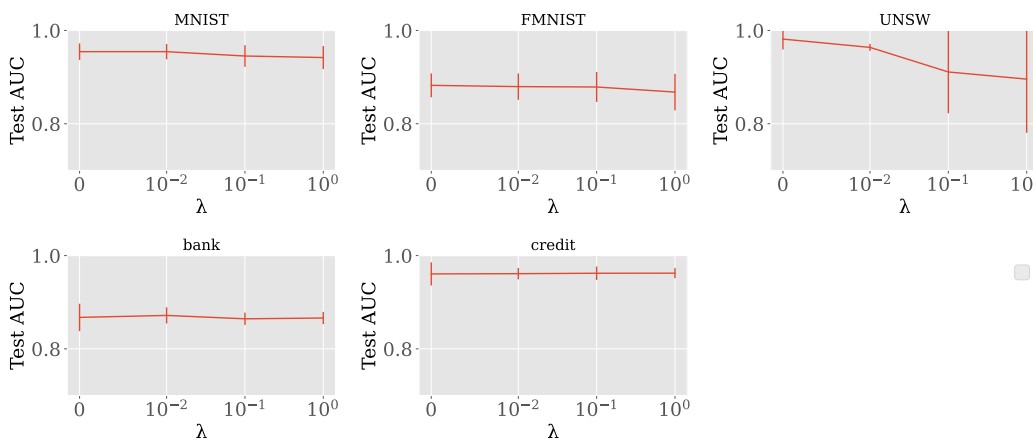

**Figure 3:** Test AUCs on the test dataset averaged over 10 runs after training on the final task **when $\lambda$ changes**.

### B.2 DEPENDENCY OF THE REGULARIZATION WEIGHT $\beta$ FOR THE LOSS FUNCTION

Figure 2 shows the average and standard deviation of AUCs when the regularization weight $\beta$ changes. The figure shows that the optimal value of $\beta$ depends on the dataset. For the range of $\beta$ in this experiment, we found that the higher the beta, the better the overall performance.

### B.3 DEPENDENCY OF THE REGULARIZATION WEIGHT $\lambda$ FOR THE LATENT VECTOR OPTIMIZATION

Figure 3 shows the average and standard deviation of AUCs when the hyperparameter $\lambda$ changes in the latent vector optimization. The figure results show that the performance of the proposed method is not sensitive to $\lambda$ on the datasets except for UNSW. In this experiment, we can see that the average AUC of the proposed method is generally high when $\lambda$ is small. However, when $\lambda = 0$, the standard deviation of the AUC becomes large, indicating that a small but non-zero value of $\lambda$ is better for stable performance.

### B.4 DEPENDENCY OF NUMBER OF SEEN ANOMALIES IN TRAINING

In this subsection, we show experimental results when the number of anomalies available for training is reduced. Specifically, we reduce the number of anomalies to 1/5 compared to the experiments in

**Table 5:** Test AUCs [%] averaged over 10 runs after continually training on all tasks **when the number of anomalies available for training is reduced to 1/5**.

|  |  | FMNIST | MNIST | UNSW | bank | credit |
|---|---|---|---|---|---|---|
| AE | A-GEM | **82.14±4.50** | 75.49±8.51 | 84.74±1.86 | 79.09±3.15 | **95.69±0.45** |
|  | EWC | 64.84±9.01 | 70.37±8.69 | 85.06±1.84 | 78.71±2.20 | **95.60±0.65** |
|  | LB | 60.05±8.41 | 61.42±9.52 | 80.72±2.17 | 67.76±2.58 | 94.55±0.63 |
| BC | A-GEM | 66.28±12.41 | 64.92±7.39 | **96.69±0.53** | 79.51±3.02 | 89.70±1.84 |
|  | EWC | 63.48±10.12 | 64.25±8.78 | **90.11±10.01** | **80.92±2.14** | 89.70±1.84 |
|  | LB | 58.63±10.12 | 52.66±9.88 | 67.60±12.09 | 74.07±1.71 | 89.64±1.75 |
| Deep SAD | A-GEM | 67.79±2.70 | 68.81±1.83 | 93.89±2.27 | 73.75±4.27 | 93.11±1.36 |
|  | EWC | 61.97±6.63 | 63.43±5.97 | 93.10±2.03 | 75.26±5.39 | 91.69±2.80 |
|  | LB | 59.08±10.57 | 56.68±4.56 | 89.31±3.32 | 70.61±6.83 | 94.51±0.94 |
| VAE | A-GEM | 66.14±8.76 | 76.56±10.43 | 81.11±0.38 | 75.95±1.04 | 94.62±0.64 |
|  | EWC | 60.45±10.29 | 61.10±9.63 | 81.13±0.61 | 76.35±0.99 | 94.62±0.63 |
|  | GR | 69.58±8.55 | **83.07±10.68** | 78.82±0.66 | 76.12±1.12 | 94.61±0.65 |
|  | LB | 60.39±10.62 | 59.77±10.32 | 80.30±0.54 | 70.98±2.09 | 94.55±0.71 |
| DevNet | A-GEM | 50.80±8.44 | 46.67±9.42 | 91.30±4.14 | 67.33±6.52 | 63.30±21.54 |
|  | EWC | 59.05±9.61 | 56.51±7.51 | 92.88±2.15 | 67.67±3.13 | 68.53±29.25 |
|  | LB | 58.13±14.09 | 49.78±8.25 | 83.02±8.36 | 56.59±6.22 | 53.65±23.92 |
| BGAD | A-GEM | 75.23±7.29 | 68.04±8.18 | **94.81±3.42** | 76.90±1.64 | **95.58±0.44** |
|  | EWC | 68.81±7.54 | 61.33±8.54 | 93.70±3.58 | 79.00±1.62 | **95.51±0.52** |
|  | LB | 63.77±7.55 | 58.03±8.88 | 90.78±5.47 | 71.48±2.10 | 95.42±0.56 |
| Ours |  | **83.61±3.79** | **89.37±6.09** | **96.56±0.83** | **82.36±0.97** | 94.75±0.69 |

---

**Algorithm 2** Traning procedure of the proposed method

**Require:** Datasets $\mathcal{D}_1, \mathcal{D}_2, \ldots, \mathcal{D}_T$, batch size $B$, number of epochs $N_e$, number of normal instances to be generated $N_{no}$, number of anomalous instances to be generated $N_{ano}$

**Ensure:** Parameters $\theta^{(T)}$ and $\phi^{(T)}$ after training on all tasks

1: **for** $t = 1 \ldots T$ **do**
2:     **if** $t \geq 2$ **then**
3:         $\mathcal{D}_{ano} = \{\boldsymbol{x}_{ano,i}\}_{i=1}^{N_{ano}} \leftarrow \texttt{GenerateAnomalies}(\theta^{(t-1)}, \phi^{(t-1)}, N_{ano})$
4:         $\mathcal{D}_t \leftarrow \mathcal{D}_t \cup \mathcal{D}_{ano}$
5:         $(\theta^{(t)}, \phi^{(t)}) \leftarrow (\theta^{(t-1)}, \phi^{(t-1)})$
6:     **end if**
7:     **for** $e = 1 \ldots N_e$ **do**
8:         $N_b \leftarrow \lceil |\mathcal{D}_t|/B \rceil$                            ▷ Number of iterations for each epoch
9:         **for** $b = 1 \ldots N_b$ **do**
10:             **if** $t \geq 2$ **then**
11:                 $\mathcal{B}_{no} = \{\boldsymbol{x}_{no,i}\}_{i=1}^{\lceil N_{no}/B \rceil} \sim p_{\theta^{(t-1)}}(\boldsymbol{x} \mid \boldsymbol{z}, y = 0)$
12:             **else**
13:                 $\mathcal{B}_{no} \leftarrow \emptyset$
14:             **end if**
15:             $\mathcal{B}_b \leftarrow \texttt{RetrieveBatch}(\mathcal{D}_t, b)$
16:             $\mathcal{B}_{con} \leftarrow \mathcal{B}_b \cup \mathcal{B}_{no}$
17:             $(\theta^{(t)}, \phi^{(t)}) \leftarrow \texttt{Update}(\theta^{(t)}, \phi^{(t)}, \mathcal{B}_{con})$
18:         **end for**
19:     **end for**
20: **end for**

Section 5. Table 5 shows the experimental results. Experimental results show that the proposed method performs best on all datasets except the credit dataset. Also, on the credit dataset, the proposed method performs comparably to the best. This result indicates that the proposed method can show stable performance regardless of the number of seen anomalies.

## C   TRAINING PROCEDURE

Algorithm 2 shows the training procedure of the proposed model. The algorithm takes the datasets $\mathcal{D}_1, \mathcal{D}_2, \ldots, \mathcal{D}_T$, the batch size $B$, the number of epochs $N_e$, the number of normal instances to be generated $N_{no}$, and the number of anomalous instances to be generated $N_{ano}$, and returns parameters

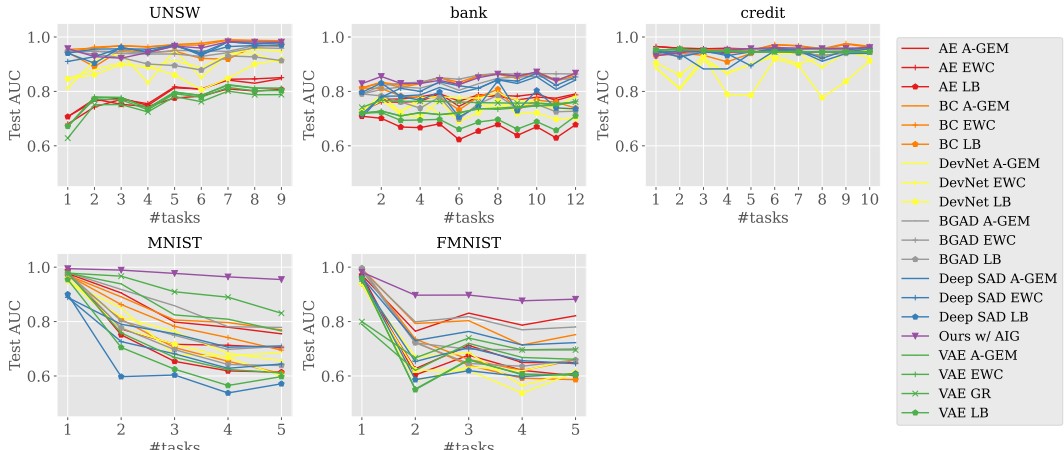

**Figure 4:** Test AUCs on seen anomalies during continually training.

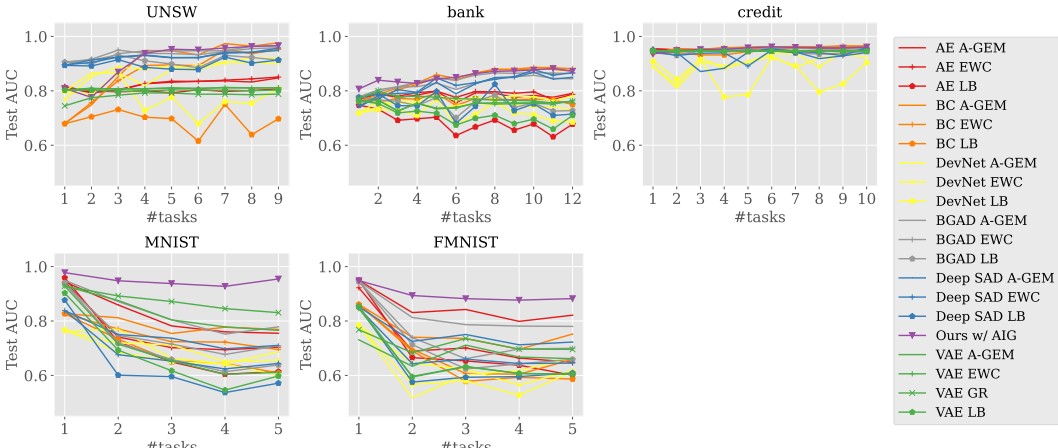

**Figure 5:** Test AUCs on seen and unseen anomalies over all the tasks during continually training.

that have been continuously trained on all tasks $\theta^{(t)}, \phi^{(t)}$. The for loop in the first line represents the part where tasks are given sequentially. In lines 3–4, if the parameters have already been trained, anomaly instances are generated in advance for the generative replay before training the current task. Here, the `GenerateAnomalies` function denotes the anomaly instance generation method described in Section 4.3. In line 5, the model parameters trained in the previous tasks are assigned to the current parameters $(\theta^{(t)}, \phi^{(t)})$. The for loop in lines 7–19 corresponds to each epoch. Line 8 calculates the number of iterations based on the batch size. The for loop in lines 9–18 corresponds to each iteration. In lines 10–14, normal instances are generated for each batch and stored to the set $\mathcal{B}_{\text{no}}$. Next, line 15 retrieves a mini-batch from the dataset $\mathcal{D}_t$. The retrieved mini-batch and the generated normal instances are concatenated in line 16. Finally, the mini-batch $\mathcal{B}_{\text{con}}$ is used to update parameters in line 17.

## D    CHANGE OF PERFORMANCE DURING CONTINUALLY TRAINING

Figures 4, 5, and 6 show the change of AUC during continually learning. Figure 4, 5, and 6 show the AUC for the seen anomalies, all anomalies across all tasks, and the unknonw (unseen) anomaies, respectively. With the exception of the early UNSW tasks, the proposed method was found to exhibit top or near top performance regardless of the datasets or tasks.

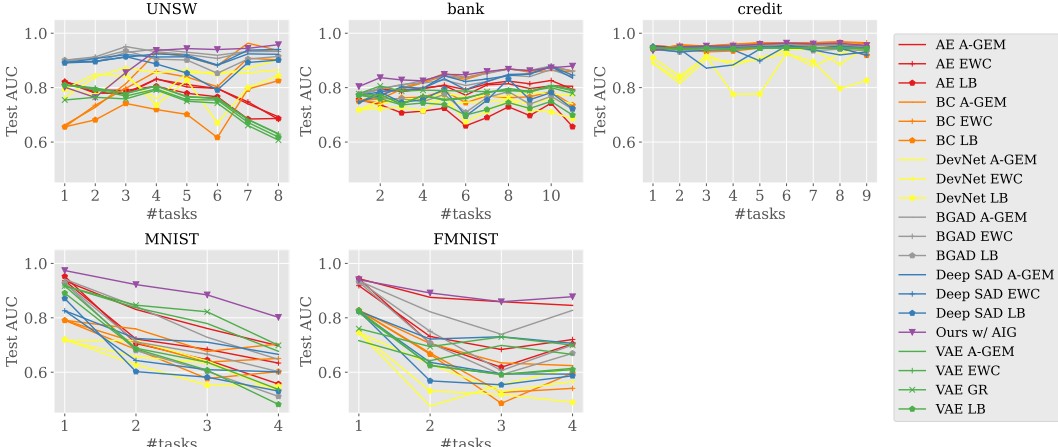

**Figure 6:** Test AUCs on unseen anomalies during continually training.

