# OpenReview forum: "Continual Supervised Anomaly Detection"
_ICLR.cc/2024/Conference — Submitted to ICLR 2024_

### Official Review · Reviewer_1EUX · 2023-11-01

**Soundness:** 3 good
**Presentation:** 3 good
**Contribution:** 2 fair
**Rating:** 6
**Confidence:** 3

**Summary:**

Many anomaly detection papers assume that only normal instances are present in the training, and train the model unsupervised.
However, in the real world, there are situations where even a few labeled abnormal instances are available.
In this case, studies have shown that even a very small number of anomalies can significantly improve the performance of the detector.
In addition, anomaly detectors are often trained under the assumption that the data distribution is stationary, but in real-world deployments, the distribution changes over time.
Therefore, the authors propose a supervised anomaly detection method using continual learning.
The method consists of a Variational AutoEncoder (VAE) and a binary classifier.
The VAE uses the reconstruction error to determine whether the input data is an unseen anomaly, and the binary classifier determines whether the input data is a seen anomaly, and calculates the anomaly score by aggregating the results of both models.
In addition, the VAE's decoder is used to generate data, which is then used for generative replay to prevent catastrophic forgetting in continual learning.

**Strengths:**

- The paper is well organized and the notation is easy to follow.
- The structure of the model and the organization of the methods (such as loss) are theoretically clean and natural. The authors naturally integrated supervised anomaly detection with continual learning.
- The proposed method works with various types of input data, such as images and tabular data.
- It is impressive that the method utilizes CVAE and a binary classifier to learn the process of generating rare abnormal instances by gradient descent, which is then used for generative replay.

**Weaknesses:**

The main weakness is that experimental results do not sufficiently support the superiority of this method.

- On tabular datasets such as UNSW, bank, and credit, the model does not significantly outperform the other baselines. In many cases, the performance is similar to that of the binary classifier, suggesting that the performance is due to the binary classifier included in the method rather than the proposed method.
- The experimental baselines are too simple. BC and VAE are components of the proposed method, and there are many methods that might outperform DevNet and Deep SAD, at least in the image domain (Liu et al., 2023). Many anomaly detection methods in the image domain are not designed for continual learning, but since EWC and A-GEM can be applied, it would be meaningful if the proposed method outperforms in this setting.
- In the image domain, the proposed method shows better performance than other baselines, but it seems that experiments on larger datasets are needed to show the practicality of the proposed method. The method was only tested on FMNIST and MNIST with MLP structure, but it would be useful to test it on larger datasets such as CIFAR10 and CelebA.
---
**Liu et al.** [Deep Industrial Image Anomaly Detection: A Survey](https://arxiv.org/abs/2301.11514). *arXiv*, 2023

**Questions:**

- Similar to other continuous-learning papers, it would be nice to be able to see how performance changes with additional training on each task.

---

> ### Author Response · Authors · 2023-11-18
> **Reply to 1EUX**
>
> Thank you for your valuable review and suggestions.
>
>
> - On tabular datasets such as UNSW, bank, and credit, the model does not significantly outperform the other baselines. In many cases, the performance is similar to that of the binary classifier, suggesting that the performance is due to the binary classifier included in the method rather than the proposed method.
>
> As you mentioned, the difference between the performances of our method and the binary classifier is very small in Table 1. However, the performance of the binary classifier should decrease when the number of anomalies available for training becomes small because the binary classifiers are not suitable for detecting unseen anomalies. To confirm this, we conducted some experiments when the number of anomalous instances available for training was reduced to 1/5. The following are the results.
>
> |                       | FMNIST          | MNIST           | UNSW            | bank           | credit          |
> |:----------------------|:----------------|:----------------|:----------------|:---------------|:----------------|
> | BC, A-GEM       | 66.28$\pm$12.41 | 64.92$\pm$7.39  | **96.69$\pm$0.53**  | 79.51$\pm$3.02 | 89.70$\pm$1.84  |
> | BC, EWC         | 63.48$\pm$10.12 | 64.25$\pm$8.78  | **90.11$\pm$10.01** | **80.92$\pm$2.14** | 89.70$\pm$1.84  |
> | BC, LB          | 58.63$\pm$10.12 | 52.66$\pm$9.88  | 67.60$\pm$12.09 | 74.07$\pm$1.71 | 89.64$\pm$1.75  |
> | Ours           | **83.61$\pm$3.79**  | **89.37$\pm$6.09**  | **96.56$\pm$0.83**  | **82.36$\pm$0.97** | **94.75$\pm$0.69** |
>
> We can see that while the performance of the binary classifier has dropped significantly, the performance of the proposed method has not dropped that much. The difference is especially obvious for the credit dataset, which is tabular data.
>
> - Many anomaly detection methods in the image domain are not designed for continual learning, but since EWC and A-GEM can be applied, it would be meaningful if the proposed method outperforms in this setting.
>
> To compare our method with the latest supervised anomaly detection method, we have implemented BGAD[3], mentioned by Reviewer QxJr, and conducted experiments. The experimental results are shown below.
> [3] Yao X, Li R, Zhang J, et al. Explicit Boundary Guided Semi-Push-Pull Contrastive Learning for Supervised Anomaly Detection. CVPR, 2023.
>
> |                       | FMNIST          | MNIST           | UNSW            | bank            | credit         |
> |:----------------------|:----------------|:----------------|:----------------|:----------------|:---------------|
> | BGAD, A-GEM     | 78.01$\pm$4.98  | 77.85$\pm$6.15  | 96.52$\pm$2.22  | **84.41$\pm$2.05** | **95.52$\pm$0.89** |
> | BGAD, EWC       | 69.98$\pm$11.63 | 70.91$\pm$8.40  | 95.83$\pm$1.75  | **86.40$\pm$1.80** | **95.66$\pm$0.97** |
> | BGAD, LB        | 65.91$\pm$7.16  | 63.85$\pm$7.76  | 91.33$\pm$6.33  | 72.74$\pm$4.00 | 94.39$\pm$1.93 |
> | Ours        | **88.23$\pm$2.56**  | **95.46$\pm$1.64**  | **98.15$\pm$2.21**  | **86.75$\pm$2.93** | **96.12$\pm$1.22** |
>
> Although the method uses the trained backbone model to extract the feature maps from the inputs, it was removed in the experiment above because it does not fit our paper's problem settings. This is because one of the goals of this paper is how to continuously train models from scratch from multiple tasks consisting of a small number of data, and the availability of models trained on large data sets in advance, such as backbone models, is inconsistent with this goal. The hyperparameters for BGAD basically followed its paper and GitHub implementation. We have added the details of the experiments to our manuscript. From the above results, we can see that our method outperforms the latest supervised anomaly detection methods.
>
> - In the image domain, the proposed method shows better performance than other baselines, but it seems that experiments on larger datasets are needed to show the practicality of the proposed method. The method was only tested on FMNIST and MNIST with MLP structure, but it would be useful to test it on larger datasets such as CIFAR10 and CelebA.
>
> As we mentioned in the replay to Reviewer vozw, our method is not domain-specific and is applicable to other domains, including tabular data. We conducted experiments on MNIST and FMNIST to show that the proposed method can detect anomalies even in image domains, and we believe these results are helpful enough to show that. Of course, using the proposed method on difficult image datasets would also be possible by using powerful generative models (VQ-VAE, GAN, diffusion models, etc.) instead of VAE.  A more in-depth investigation of proposed methods for specific domains is a topic for future work.
>
> - Similar to other continuous-learning papers, it would be nice to be able to see how performance changes with additional training on each task.
>
> Thank you for your advice. We will add to the paper the performance changes with additional training on each task.

---

> > ### Comment · Reviewer_1EUX · 2023-11-21
> >
> > Thanks to the authors for the answers and additional experiments. I think the proposed approach is interesting and will be helpful for future research, and I decided to raise the review score. But I still have concerns about its practicality and performance on large datasets.

---

> > > ### Author Response · Authors · 2023-11-22
> > >
> > > Thank you for your response and for your willingness to update your score. The discussion period will be ending shortly, but we would be happy to answer your questions about large datasets if the experiment is completed during the discussion period.

---

### Official Review · Reviewer_QxJr · 2023-11-02

**Soundness:** 3 good
**Presentation:** 3 good
**Contribution:** 2 fair
**Rating:** 5
**Confidence:** 4

**Summary:**

This paper proposes a new approach to the task of continual supervised anomaly detection. This paper designs a pipeline with three specific components: a variational autoencoder, a binary classifier, and an anomaly generation mechanism. This paper conducts experiments on five datasets to validate performance.

**Strengths:**

•	This paper proposes a new pipeline for continual supervised anomaly detection that aligns well with the practical application needs.
•	 The performance gains for AUC on five datasets look good, especially on the FMNIST and MNIST datasets.

**Weaknesses:**

•	The novelty of the proposed framework is limited. The overall network architecture consists of a VAE and a classifier without any particularly unique components.

•	This paper does not include a comparison with some of the latest supervised anomaly detection methods such as DRA[1], PRN[2], BGAD[3], which might be relevant for a more comprehensive evaluation.

[1] Ding C, Pang G, Shen C. Catching both gray and black swans: Open-set supervised anomaly detection. CVPR 2022.

[2] Zhang H, Wu Z, Wang Z, et al. Prototypical residual networks for anomaly detection and localization. CVPR 2023.

[3] Yao X, Li R, Zhang J, et al. Explicit Boundary Guided Semi-Push-Pull Contrastive Learning for Supervised Anomaly Detection. CVPR, 2023.

**Questions:**

Is there a more detailed explanation regarding the impact of the number of seen anomalous samples on the experimental results?

---

> ### Author Response · Authors · 2023-11-18
> **Reply to Reviewer QxJr**
>
> We thank you for reading our paper and your review comments.
> - The novelty of the proposed framework is limited.
>
> As you mentioned, our method consists of a binary classifier and a VAE, which are very simple and basic modules. However, this does not mean that the novelty of the proposed method is limited. In fact, it is not obvious how to combine VAE with a binary classifier and how to perform generative replay from them. Our method's novelty is not in using VAEs or binary classifiers but in finding a way to use them to solve the problem of continual supervised anomaly detection. For example, in Eqs. (7)-(9), the formulation of generating anomalies is naturally derived by appropriately combining VAE and a binary classifier. In addition, its simplicity has the advantage that it can be easily extended if necessary (e.g., by changing the generative model or the classifier).
>
> - This paper does not include a comparison with some of the latest supervised anomaly detection methods such as DRA[1], PRN[2], BGAD[3], which might be relevant for a more comprehensive evaluation.
>
> Thank you for your comment. For a more comprehensive evaluation, we have conducted the experiments with BGAD[3]. The following are the results.
>
> |                       | FMNIST          | MNIST           | UNSW            | bank            | credit         |
> |:----------------------|:----------------|:----------------|:----------------|:----------------|:---------------|
> | BGAD, A-GEM     | 78.01$\pm$4.98  | 77.85$\pm$6.15  | 96.52$\pm$2.22  | **84.41$\pm$2.05** | **95.52$\pm$0.89** |
> | BGAD, EWC       | 69.98$\pm$11.63 | 70.91$\pm$8.40  | 95.83$\pm$1.75  | **86.40$\pm$1.80** | **95.66$\pm$0.97** |
> | BGAD, LB        | 65.91$\pm$7.16  | 63.85$\pm$7.76  | 91.33$\pm$6.33  | 72.74$\pm$4.00 | 94.39$\pm$1.93 |
> | Ours        | **88.23$\pm$2.56**  | **95.46$\pm$1.64**  | **98.15$\pm$2.21**  | **86.75$\pm$2.93** | **96.12$\pm$1.22** |
>
> Although the methods you mentioned [1-3] rely on the trained backbone model to extract the feature maps from the inputs, it was removed in the experiment above because it does not fit our paper's problem settings. This is because one of the goals of this paper is how to continually train models from scratch using multiple tasks consisting of a small number of data, and the availability of models trained on large datasets in advance, such as backbone models, is inconsistent with this goal. The hyperparameters for BGAD basically followed its paper and GitHub implementation. We have added the details of the experiments to our manuscript. From the above results, we can see that our method outperforms the latest supervised anomaly detection methods.
>
> - Is there a more detailed explanation regarding the impact of the number of seen anomalous samples on the experimental results?
>
> Thank you for your question. To investigate the impact of the number of seen anomalous instances to be used for training, we conducted some experiments when the number of seen anomalous instances available for training was reduced to 1/5. The following are the results.
>
> |                       | FMNIST          | MNIST           | UNSW            | bank           | credit          |
> |:----------------------|:----------------|:----------------|:----------------|:---------------|:----------------|
> | BC, A-GEM       | 66.28$\pm$12.41 | 64.92$\pm$7.39  | **96.69$\pm$0.53**  | 79.51$\pm$3.02 | 89.70$\pm$1.84  |
> | BC, EWC         | 63.48$\pm$10.12 | 64.25$\pm$8.78  | **90.11$\pm$10.01** | **80.92$\pm$2.14** | 89.70$\pm$1.84  |
> | Deep SAD, A-GEM | 67.79$\pm$2.70  | 68.81$\pm$1.83  | 93.89$\pm$2.27  | 73.75$\pm$4.27 | 93.11$\pm$1.36  |
> | Deep SAD, EWC   | 61.97$\pm$6.63  | 63.43$\pm$5.97  | 93.10$\pm$2.03  | 75.26$\pm$5.39 | 91.69$\pm$2.80  |
> | DevNet, A-GEM   | 50.80$\pm$8.44  | 46.67$\pm$9.42  | 91.30$\pm$4.14  | 67.33$\pm$6.52 | 63.30$\pm$21.54 |
> | DevNet, EWC     | 59.05$\pm$9.61  | 56.51$\pm$7.51  | 92.88$\pm$2.15  | 67.67$\pm$3.13 | 68.53$\pm$29.25 |
> | BGAD, A-GEM     | 75.23$\pm$7.29  | 68.04$\pm$8.18  | **94.81$\pm$3.42**  | 76.90$\pm$1.64 | **95.58$\pm$0.44**  |
> | BGAD, EWC       | 68.81$\pm$7.54  | 61.33$\pm$8.54  | 93.70$\pm$3.58  | 79.00$\pm$1.62 | **95.51$\pm$0.52**  |
> | Ours           | **83.61$\pm$3.79**  | **89.37$\pm$6.09**  | **96.56$\pm$0.83**  | **82.36$\pm$0.97** | 94.75$\pm$0.69  |
>
> We show only the results for the supervised anomaly detection methods because the performances of the unsupervised anomaly detection methods (i.e., AE and VAE) do not depend on it. The complete table has been added to the manuscript. Note that in the experiment, only in the credit dataset, the performance of the proposed method is not at the top, but the difference between the performances of the top methods and ours is still small enough. The table shows that our method shows stable performance regardless of the number of anomalous instances for training.

---

> > ### Author Response · Authors · 2023-11-22
> > **Gentle Reminder for Discussion to Reviewer QxJr**
> >
> > Thank you again for your constructive and valuable comments. We would like to remind you that the discussion period ends on November 22. Did our additional experiments address your concerns? If you have any other comments or questions, please let us know.

---

> > ### Comment · Reviewer_QxJr · 2023-11-22
> >
> > Thanks to the authors for their responses and additional experiments, which addressed some of my concerns. However, I still think the novelty of the overall architecture is limited and the improvement on tabular datasets seems limited. And, since significant gains are mainly observed on image datasets, I believe that validating this approach on complex and large image datasets is a necessity rather than an alternative.   Based on these considerations, I maintain my review score unchanged.

---

### Official Review · Reviewer_vozw · 2023-11-04

**Soundness:** 3 good
**Presentation:** 3 good
**Contribution:** 2 fair
**Rating:** 6
**Confidence:** 3

**Summary:**

The authors propose a method for semi-supervised anomaly detection in the setting of continuous learning. They combine a binary classifier for the labeled anomalies together with a VAE reconstruction score for the anomaly detection part. The continuous learning is addressed by, one, using the latent space of the VAE to sample data from past tasks, and, two, using the gradient of the binary classifier to sample labeled anomalies in the same latent space via iteration from a starting point.

**Strengths:**

The paper is easily readable.
They perform experiments on various types of datasets.
They perform an ablation study.

**Weaknesses:**

It represents a straightforward combination of ideas.
The argument that one cannot keep data due to privacy reasons also applies for resampling data from an autoencoder. If it is a very good reconstruction, it would equally cause privacy issues.
The important and relevant case of slow distribution shift is only partially addressed, via the credit data. Doing so in more and more controllable settings would be of interest.
A comparison against reusing past training data would be of interest - in particular from tasks a few epochs ago. Reason being that a slow shift of parameters would also affect sampling of data from not so recent past tasks.
MNIST and FMNIST might be too simple as problem. A more complex image dataset is missing.

**Questions:**

NA

---

> ### Author Response · Authors · 2023-11-18
> **Reply to Reviewer vozw**
>
> We sincerely appreciate your elaborate reading of our paper and your insightful comments.
>
> - It represents a straightforward combination of ideas.
>
> As you mentioned, our method looks very simple, but it does not mean the lack of novelty of our method. For example, the formulation for generating anomaly instances from VAEs is not a mere heuristic, but is derived naturally from the structure of our model. This was obtained by appropriately combining VAEs and binary classifiers to create our model. In fact, our method outperforms the conventional methods.
>
> - The argument that one cannot keep data due to privacy reasons also applies for resampling data from an autoencoder. If it is a very good reconstruction, it would equally cause privacy issues.
>
> Yes, the generator could cause privacy issues if the generated instances are of high quality. However, the risk should be reduced compared to keeping the data directly. For example, the webpage of SyntheticData4ML workshop says that "Synthetic data is regarded as one potential way to promote privacy. The 2019 NeurIPS Competition "Synthetic data hide and seek challenge" demonstrates the difficulty in performing privacy attacks on synthetic data." Further investigation into this issue is planned for future work.
> https://www.syntheticdata4ml.vanderschaar-lab.com/
>
> - The important and relevant case of slow distribution shift is only partially addressed, via the credit data. Doing so in more and more controllable settings would be of interest.
>
> Many previous studies on continual learning focused on mitigating catastrophic forgetting, and thus, they are more interested in quickly changing data distributions than in slowly changing ones. This is because catastrophic forgetting is unlikely to occur when the shift in distribution is small. However, as you mentioned, it is important to understand how well the proposed method works in a problem setting where the distribution changes gradually because such situations can occur in real-world applications. In this paper, we investigate such a situation on a credit dataset and confirm that our method is superior to the others. We believe that the superiority of the proposed method will probably remain unchanged in such situations where the distribution change is small.
>
> - A comparison against reusing past training data would be of interest - in particular from tasks a few epochs ago. Reason being that a slow shift of parameters would also affect sampling of data from not so recent past tasks.
>
> Thank you for your suggestion. The situation in which data from previous tasks can be used differs from the problem settings in this paper and the usual continual learning studies, making the problem much easier. Such a problem setting would be interesting, but may be outside the scope of this paper.
>
> - MNIST and FMNIST might be too simple as problem. A more complex image dataset is missing.
>
> Please note that our proposed method is not domain-specific, although its application to more challenging image datasets is important. Our approach can be applied to non-image datasets such as tabular data. We have included experimental results on MNIST and FMNIST to confirm that our proposed method can also be used in the image domain, and we believe that they played a sufficient role in this regard. Of course, it would also be possible to use the proposed method on difficult image datasets by using powerful generative models (VQ-VAE, GAN, diffusion models, etc.) instead of VAE. Such considerations are the subject of future work.

---

> > ### Author Response · Authors · 2023-11-22
> > **Gentle Reminder for Discussion to Reviewer vozw**
> >
> > Thank you again for your constructive comments on our paper. We would like to remind you that the discussion period ends on November 22. We hope that our responses have correctly addressed your questions and concerns. If we have misinterpreted your comments or you have other concerns, we look forward to hearing from you.

---

> > > ### Comment · Reviewer_vozw · 2023-11-23
> > > **Comments not well adressed.**
> > >
> > > The study is more empiricial than theoretical.
> > >
> > > The message of the comments in
> > >
> > > The argument that one cannot keep data due to privacy reasons also applies for resampling data from an autoencoder. If it is a very good reconstruction, it would equally cause privacy issues.
> > >
> > > The important and relevant case of slow distribution shift is only partially addressed, via the credit data. Doing so in more and more controllable settings would be of interest.
> > >
> > > A comparison against reusing past training data would be of interest - in particular from tasks a few epochs ago. Reason being that a slow shift of parameters would also affect sampling of data from not so recent past tasks.
> > >
> > > is that from a practical viewpoint it would be useful to evaluate it, even more so as the paper is focused on empirical results.
> > > After deployment a slow shift is more likely than a strong task shift ( exceptions being maybe in cyber attacks).
> > >
> > > That other studies have not done it, is not a convincing argument.
> > >
> > >
> > > Please note that our proposed method is not domain-specific, although its application to more challenging image datasets is important. Our approach can be applied to non-image datasets such as tabular data. We have included experimental results on MNIST and FMNIST to confirm that our proposed method can also be used in the image domain, and we believe that they played a sufficient role in this regard.
> > >
> > > That line of thought ignores the problem, that on more realistic datasets, methods might behave very differently in comparison.
> > >
> > > The reviewer feels that the comments are not satisfactorily adressed and keeps the current score.

---

### Author Response · Authors · 2023-11-23

Thank you again for your valuable and helpful feedback. The main concern of all the reviewers was whether the proposed method would be effective on large datasets. Although the discussion will be over soon, we have performed continual anomaly detection on the CelebA dataset and would like to share the results with you.

Due to time constraints, we compared four supervised anomaly detection methods: binary classifier, DevNet, BGAD, and the proposed method, and used A-GEM and EWC as continual learning methods. The resolution of the CelebA dataset was reduced to 32 x 32 pixels in this experiment because of its large resolution. In addition, we performed five runs instead of 10 for each condition and evaluated the results using the average AUC.

To create tasks, we first selected the five attributes with the smallest number of data ("Bald," "Double_Chin," "Pale_Skin," "Gray_Hair," and "Mustache") among the attributes of the CelebA dataset. We then defined each attribute as an anomaly class. Image data with more than one of these attributes were deleted. Five tasks were created by dividing the normal data into five parts and assigning an anomaly class to each part.

The model architecture used was the one provided in the GitHub implementation at the URL below.

https://github.com/podgorskiy/VAE

The $\lambda$ and $\beta$ in the proposed method were determined in advance by conducting image generation experiments using CelebA (i.e., $\lambda=0.01$, $\beta=50$). Since we found that it is better to vary the learning rate for the CelebA dataset, we selected learning rates from {1e-3, 1e-4, ..., 1e-5} for all the methods according to the validation set. We also modified our anomaly score function defined in Eq.(4) by introducing a hyperparameter $\alpha$ that controls whether seen or unseen anomalies are detected more strongly (i.e., $s_{\theta, \phi}(\mathbf{x}) := -\alpha \mathbb{E}_{q\_\phi(\mathbf{z}\mid \mathbf{x})} \log p\_\theta(\mathbf{x}\mid \mathbf{z}, y=0) + (1-\alpha)\log r\_\phi(y=1\mid \mathbf{x})$). $\alpha$ was also chosen from {1e2, ..., 1e-2} using the validation set.



The results of the experiment are shown below.
|                       | CelebA       |
|:----------------------|:---------------|
| BC, A-GEM       | 72.44$\pm$3.20 |
| BC, EWC         | 69.68$\pm$2.74 |
| DevNet, A-GEM   | 70.44$\pm$1.06 |
| DevNet, EWC     | 69.76$\pm$0.45 |
| BGAD, A-GEM     | 71.20$\pm$2.59 |
| BGAD, EWC       | 71.15$\pm$3.07 |
| Ours          | 73.00$\pm$3.68 |

As shown in the experimental results, the proposed method performed the best on average. However, the performance of all methods was relatively close. This may be due to the large number of anomalous data avaiable for training in this experiment (from 5% to 10%), and the difference between the performances of the proposed method and the binary classifiers may increase when the number of anomalous data is further reduced.

---

> ### Comment · Reviewer_vozw · 2023-11-23
>
> The reviewer recognizes this result. However with 32x32 one has the same limited spatial complexity as for mnist/cifar-10/100 . with 32x32 the complexity of it is largely gone.

---

> > ### Author Response · Authors · 2023-11-23
> >
> > Thank you for your prompt reply. Indeed, the image size of 32 x 32 may be small, but the complexity of the dataset should be higher than MNIST or FMNIST. Also, Reviewer 1EUX recommends testing the proposed method with CIFAR10 as well as CelebA as a larger dataset.

---

### Meta-Review · Area_Chair_iNVb · 2023-12-06

**Metareview:**

The manuscript presents a continual learning model for anomaly detection with few labelled samples. The continual learning is enabled through a clever combination of generative replay using VAE and a binary classifier.

Strength:
1. The proposed method is evaluated on multiple modalities (image and structured data) [Reviewer vozw, QxJr, 1EUX]
2. It is impressive that the method utilizes CVAE and a binary classifier to learn the process of generating rare abnormal instances by gradient descent, which is then used for generative replay. [Reviewer 1EUX]

Weakness:
1. The proposed method is a combination of existing approaches (VAE and binary classifier) and lacks novelty [Reviewer vozw, QxJr]
2. The evaluations are insufficient - 2 very simple image data sets and a few tabular data sets. It is also not evident why these modalities were chosen (for eg., why not text data sets? Why not time-series data sets?). Another way could have been to restrict to a single modality and demonstrate the algorithm on a few complex data sets in that modality.

**Justification For Why Not Higher Score:**

The  manuscript lacks a rigor in evaluation.

**Justification For Why Not Lower Score:**

N/A

---

### Decision · Program_Chairs · 2024-01-16

Reject